# Pheromones in Crane Flies: Behaviorally Active Cuticular Compounds in *Tipula autumnalis* Loew (Diptera: Tipulidae)

**DOI:** 10.3390/insects16010024

**Published:** 2024-12-29

**Authors:** Vincas Būda, Violeta Apšegaitė, Laima Blažytė-Čereškienė, Sigitas Podėnas, João Pedro de A. Souza, Paulo H. G. Zarbin, Linas Labanauskas, Tomas Paškevičius, Vilma Baužienė, Sandra Radžiutė

**Affiliations:** 1Nature Research Centre, Akademijos St. 2, LT-08412 Vilnius, Lithuania; violeta.apsegaite@gamtc.lt (V.A.); laima.blazyte@gamtc.lt (L.B.-Č.); sigitas.podenas@gamtc.lt (S.P.); sandra.radziute@gamtc.lt (S.R.); 2Departamento de Quimica, Laboratorio de Semioquimicos, Universidade Federal do Paranȧ, Curitiba 81531-990, PR, Brazil; albuquerquesjp@gmail.com (J.P.d.A.S.); pzarbin@ufpr.br (P.H.G.Z.); 3Department of Organic Chemistry, Institute of Chemistry and Chemical Technology, Center for Physical Sciences and Technology, Sauletekio Av. 3, LT-10257 Vilnius, Lithuania; linas.labanauskas@ftmc.lt (L.L.); tomas.paskevicius@ftmc.lt (T.P.)

**Keywords:** cuticular hydrocarbons, olfactory response, GC-EAD, sex pheromone, mate recognition

## Abstract

The role of pheromones in crane flies (family Tipulidae) has never been studied before. This family contains over 15,000 species, including two recognized as pests. Pheromones are an important tool for controlling insect behavior. Therefore, the aim of this study was to determine whether crane flies produce pheromones, and if so, to identify the chemical compounds involved. To achieve this, we selected a model species with features that would make pheromone detection easier. *Tipula autumnalis* was chosen because males have well-developed antennae, which are essential for detecting smells, and females are wingless, requiring them to attract males. Using advanced techniques, we identified nine key chemical compounds, all classified as cuticular hydrocarbons, which triggered a response in males. In behavioral tests, three compounds—*n*-pentacosane, (*Z*)-9-pentacosene, and (*Z*, *E*)-6,9-pentacosadiene—attracted males, suggesting that these are components of the female sex pheromone. Some evidence also suggests that a few of these compounds may play a role in male–male interactions. Chemical communication in *T. autumnalis* is therefore quite complex.

## 1. Introduction

Insect cuticular hydrocarbons (CHCs) serve not only as a water barrier but also play an important role in intraspecific communication, including recognition, aggregation, dispersal, alarm, and sexual behavior, in many insect species [1]. The role of CHCs as sex pheromones is well established in insects across several orders, including Coleoptera, Diptera, Hemiptera, Hymenoptera, and Thysanoptera [2]. Among dipterans, certain CHCs have been identified as sex pheromones in representatives of six families within the Brachycera-Cyclorrhapha suborder (Agromyzidae, Drosophilidae, Fanniidae, Glossinidae, Muscidae, Oestridae) and two families within the Nematocera suborder (Ceratopogonidae, Sciaridae) [1,3]. For instance, it has been reported that female *Tipula paludosa* stimulate mating attempts in males from a short distance, though it remains unclear whether visual or chemical stimuli are responsible for this effect [4].

Although the Tipulidae are one of the largest groups within the Diptera, containing over 15,270 valid species and subspecies [5], no data on chemical communication are available for this family. Two species within the family are considered significant agricultural pests in Europe and North America, namely the common European crane fly (*T. paludosa* Meigen, 1830) and the marsh crane fly (*T. oleracea* Linnaeus, 1758) [6]. The larvae of these crane fly species feed on the roots of spring cereals, brassica crops, potatoes, maize, lawn grasses, and clover. Additionally, *T. paludosa* has been reported as an insect pest, causing damage to turfgrass on golf courses [7].

For this study, we selected *Tipula autumnalis* Loew, 1864 (Diptera: Tipulidae) as a model species that we found to be most suitable for our research. This species was chosen because males have well-developed antennae, while females possess reduced antennae. Additionally, females have short, reduced wings and are unable to fly. *T. autumnalis* is a univoltine species, and its larvae feed on plant roots. Our hypothesis was that *T. autumnalis* virgin females release sex pheromones. This prediction arose from observations of female behavior, which included a stationary posture characterized by abdominal elevation. In this posture, the terminal abdominal segments and cerci were slightly elevated, and the abdomen moved slowly up and down. Under laboratory conditions, *T. autumnalis* males copulated or attempted to copulate only with females that exhibited this specific behavior.

The aim of our study was to determine whether *T. autumnalis* females release chemical compounds involved in mate recognition and mating. We also sought to identify electrophysiologically and behaviorally active compounds that could function as sex pheromone components.

## 2. Materials and Methods

### 2.1. Insects

*Tipula autumnalis* adults were collected from September to October 2014–2017 in two natural habitats in Lithuania, namely Dūkštai oak forest (E24.963667, N54.838604), Vilnius district, and Būda forest (E24.35851, N54.88575), Kaišiadorys district. The species was identified based on its morphology by S. Podėnas. *T. autumnalis* voucher specimens were lodged in the entomological collection at the Nature Research Centre (Vilnius, Lithuania). Live crane flies were maintained in individual vials (150 mL), supplied with wet filter paper, and stored at low temperature (ca. 4 °C) for 6–8 days.

### 2.2. Preparation of Cuticular Washes

The *T. autumnalis* crane flies were sexed, and groups of 30 specimens, either males or females, were used to prepare cuticular washes. Only females exhibiting specific behavior (abdominal elevation) were selected. The insects were killed by freezing, and their abdomens were cut off and washed in hexane (30 abdomens/1 mL) for 5 min. The washes were concentrated under N_2_ to ~250 µL and stored in a freezer at −18 °C. Finally, the analyses of gas chromatography–electroantennogram detection (GC-EAD) and gas chromatography–mass spectrometry (GC-MS), as well as behavioral tests, were conducted on the washes.

### 2.3. Gas Chromatography-Electroantennogram Detection

For GC-EAD analysis, 1 µL of cuticular washes was splitlessly injected into a Clarus 500 gas chromatograph (PerkinElmer, Waltham, MA, USA) equipped with an DB-Wax capillary column (30 m; 0.25 mm i.d., 0.25 µm film thickness, Agilent Technologies, Santa Clara, CA, USA). A 1:1 effluent splitter allowed for simultaneous flame ionization detection (FID) and EAD of the separated compounds. Hydrogen was used as the carrier gas (1.5 mL/min). The injector and detector temperatures were set at 240 °C. The oven temperature was programmed to increase from 80 °C to 240 °C at a rate of 8 °C/min, followed by an isothermal temperature for 14 min.

A humidified airstream flowing at 0.5 m/s through the outlet for the EAD was held over antenna preparation. An indifferent electrode, a glass capillary filled with 0.9% NaCl saline (Ilsanta, Vilnius, Lithuania) and grounded via a silver wire, was inserted into the severed crane fly head with an antenna. A recording electrode connected to a high-impedance DC amplifier was connected to the cut tip of the antenna. The signal was stored and analyzed on a PC equipped with an IDAC card using the program GC-EAD V. 4.4 (Syntech, NL 1998). Eleven male and twelve female antennae were used for the GC-EAD analysis. A CHC was considered to be electrophysiologically (EAG) active if it elicited a response in at least three successful GC-EAD recordings.

### 2.4. Gas Chromatography-Mass Spectrometry

GC-MS analyses were carried out using a Shimadzu GCMS-QP2010 Plus (Shimadzu, Tokyo, Japan) equipped with a Stabilwax column (30 m; 0.25 mm i.d., 0.25 µm film thickness, Restek, Bellefonte, PA, USA) with a temperature programmed to increase from 80 °C to 250 °C at 8 °C/min and then held isothermally for 25 min. and a Rxi-1 ms column (30 m; 0.25 mm i.d., 0.25 µm film thickness, Restek, Bellefonte, PA, USA) with a temperature programmed to increase from 80 °C to 300 °C at 6 °C/min and then held isothermally for 20 min. The injector, the temperature of which was set at 240 °C, was operated in splitless mode. Helium was used as the carrier gas at a flow rate of 1.2 mL/min. Electron ionization spectra were acquired at an electron energy of 70 eV, and the ion-trap source temperature was set at 240 °C. For analysis, 1 μL of sample was injected. The compounds were identified by comparing their retention indices (RIs) and mass spectra with the National Institute of Standards and Technology (NIST) reference library and the authentic standards. The diagnostic mass spectral fragments that provide unequivocal evidence of the position of the methyl branches were documented previously in detail [8]. The standard mix of *n*-alkanes (C13–C33, Restek, Bellefonte, PA, USA) was co-injected with authentic samples to determine the retention indices (RIs) of the CHC components. Unsaturated hydrocarbons were identified by mass spectra, GC retention indices, and mass spectra of the dimethyl disulphide (DMDS) adduct [9]. Samples in hexane (50 μL) were treated with 100 μL of DMDS (CAS Nr. 624-92-0, Alfa Aesar, now Thermo Fisher Scientific, Karlsruhe, Germany) and 1–2 drops of iodine solution (60 mg iodine (Sigma-Aldrich, Darmstadt, Germany) in 1 mL of diethyl ether (Sigma-Aldrich, Darmstadt, Germany)). Reaction mixtures were left overnight at 40 °C, diluted with hexane (100 μL), washed with 5% sodium thiosulfate (100 μL), and the organic phase was dried over anhydrous Na_2_SO_4_.

### 2.5. Synthesis of Chemical Compounds

The starting materials were purchased from Sigma Aldrich (1-decyne, 1-octyne, 1-bromoheptadecane, 1-bromopentadecane, hexamethylphosphoramide (HMPA), tetrahydrofuran (THF), palladium on barium sulfate 5%, n-butyllithium 2.5 M in hexanes, quinoline). Chloroform-d (99.8% D) for NMR spectroscopy and Zeoprep 60 35–70 µm silica gel for flash chromatography were purchased from Apollo Scientific. Merck Silica gel 60 F254 plates were used for TLC tests.

NMR spectra were recorded on a Bruker Avance III spectrometer (400 MHz for ^1^H NMR, 100 MHz for ^13^C NMR). Chemical shifts δ are reported with respect to the residual solvent peak and are given in ppm.

The synthesis of (*Z*)-9-pentacosene and (*Z*)-7-pentacosene were performed using methods described by Masuda and Mori [10] and Fisher et al. [11] according to the scheme (Figure 1). 9-Pentacosyne (1) and 7-pentacosyne (3) were obtained by the alkylation of lithium salts of 1-decyne and 1-octyne with 1-bromopentadecane and 1-bromoheptadecane in the presence of hexamethylphosphoric triamide (HMPA). Catalytic hydrogenation of alkynes 1 and 3 with a poisoned palladium catalyst afforded pure cis-monoenes (*Z*)-9-pentacosene (2) and (*Z*)-7-pentacosene (4) in good yields (65–70%).

For the synthesis of (*Z*)-9-pentacosene, 9-pentacosyne (1) was first synthesized. To a stirred solution of 1-decyne (1 g, 7.23 mmol) and HMPA (5 mL) in dry THF (10 mL) was added a solution of n-butyllithium in hexanes (2.5 M, 7.23 mmol, 2.36 mL) in a dropwise manner at −40 °C under argon. The reaction mixture was stirred at −20 °C for 2 h, then cooled to −40 °C. A solution of 1-bromopentadecane (2.01 g, 7.23 mmol) in dry THF (20 mL) was added dropwise to the reaction mixture at −40 °C, allowed to warm to room temperature, stirred overnight, quenched with a saturated solution of NH_4_Cl, and extracted with hexanes. The organic layer was washed with water and dried with Na_2_SO_4_, the solvent was removed under reduced pressure to obtain 2.56 g of crude product, which was crystalized from hexanes at −20 °C to yield 1.95 g (74%) of 9-pentacosyne (1) as a colorless oil.

GC-MS: retention time of main peak was 11.360 min, area 100%. ^1^H NMR (400 MHz, CDCl3) δ 2.00–2.26 (m, 4H), 1.14–1.30 (m, 38H), 0.92 (t, J = 6.7, 6H).

7-pentacosyne (**3**) was prepared using the same methods as described before to obtain the 98% pure compound as a colorless oil in 72% yield.

GC-MS: retention time of main peak was 11.395 min, area 97.88%. ^1^H NMR (400 MHz, CDCl3) δ 2.01–2.25 (m, 4H), 1.15–1.31 (m, 38H), 0.92 (t, J = 6.7, 6H). To obtain (*Z*)-9-pentacosene (2), Pd/BaSO_4_ (120 mg) was suspended in a solution of 9-pentacosyne (1.95 g, 5.32 mmol) and quinoline (2 mL) in hexane (20 mL), a balloon of hydrogen was attached, and the reaction was monitored via GC. After completion, the reaction mixture was filtered through a short plug of silica gel and solvent was removed under vacuum. The residue was purified by silica gel flash chromatography using octane as a mobile phase to yield 1.43 g (70%) of >97% pure (*Z*)-9-pentacosene (2) as a colorless oil.

GC-MS: retention time of the main peak was 11.284 min, area 96.25%. ^1^H NMR (400 MHz, CDCl3) δ 5.34 (t, J = 4.87 Hz, 2H), 2.04 (q, J = 6.40 Hz, 4H), 1.15–1.30 (m, 38H), 0.90 (t, J = 6.7, 6H).

^13^C NMR (100 MHz, CDCl3): 14.1, 22.7, 27.2, 29.3, 29.4, 29.5, 29.6, 29.7, 29.8, 31.9, 129.9. (*Z*)-7-Pentacosene (4) was prepared using the same methods as described before to give the >96% pure compound as a colorless oil in 65% yield.

GC-MS: retention time of main peak was 11.284 min, area 96.25%.

^1^H NMR (400 MHz, CDCl3) δ 5.38 (t, J = 4.87 Hz, 2H), 2.04 (q, J = 6.40 Hz, 4H), 1.15–1.30 (m, 38H), 0.91 (t, J = 6.7, 6H). ^13^C NMR (100 MHz, CDCl3): 14.2, 22.8, 27.2, 29.3, 29.4, 29.5, 29.6, 29.7, 29.8, 31.9, 129.8. The syntheses of (*Z*,*E*)-6,9-pentacosadiene and the two enantiomers, (*R*)- and (*S*)-3-methylheneicosane, were conducted at the Laboratory of Semiochemicals, Federal University of Parana, Brazil. These results will be detailed in a separate publication (Zarbin et al., in preparation). The (*Z*,*E*)-6,9-pentacosadiene was obtained as a mixture containing the (*Z*,Z)-isomer in a ratio of 3:1. The (*R*)- and (*S*)-3-methylheneicosane were synthesized using an enantioselective approach, achieving enantiomeric excesses of 98% and >99%, respectively. *R*3Me-21Hy ${\left[\alpha \right]}_{D}^{20}$ = +1.0 (c = 0.13, CHCl_3_); *S*3Me-21Hy ${\left[\alpha \right]}_{D}^{20}$ = −0.9 (c = 0.10, CHCl_3_).

### 2.6. Behavioral Assay

For the behavioral assay, two insect cages (120 mm in diameter, 75 mm in height) were constructed. Petri dish lids served as the top and bottom of the cage, while the walls were made of polypropylene net (mesh size 4 × 4 mm). A filter paper (Whatman, 5 × 15 mm) soaked with 10 μL, either of female cuticular washes (in hexane) or solvent (hexane) as a control, was stuck on the wall 20 mm from the top in the test and control cages after solvent evaporation. For visual and tactile stimuli, a dead and washed female (washed in hexane for 10 min) was placed on the treated filter paper. Behavioral tests were conducted during photophase (from 12 p.m. till 6 p.m. mainly), at room temperature (18–19 °C), 60% RH, and air exhaust. The testing time was chosen based on the observation of *T. autumnalis* mating under laboratory conditions. Synthetic compounds (*n*-pentacosane, (*Z*)-9-pentacosene, (*Z*,*E*)-6,9-pentacosadiene, (*Z*)-9-tricosene, (*S*)- and (*R*)-3-methylheneicosane) were tested in the same way as the cuticle washes.

The behavior of *T. autumnalis* males was recorded using a scan camera (acA1300-60gm, Basler, Ahrensburg, Germany) placed 225 mm above the behavioral arena (120 mm in diameter). Within the arena, the stimulus zone was distinguished (63 mm in length, 32 mm in width) (Figure 2). Recordings started immediately after the male was introduced into the cage and continued for 40 min. Eight to twelve trials were conducted for each treatment. Males were picked up randomly and each specimen was tested only once. The videos were analyzed using EthoVision XT software (v12.0, Noldus Information Technology b.v., Wageningen, The Netherlands). The cumulative time spent in the stimulus zone, the latency to the first entrance into the stimulus zone, the number of stimulus zone visits, the distance moved, and the velocity were measured. These parameters, computed by EthoVision software, were analyzed statistically using the nonparametric Mann–Whitney U test.

## 3. Results

### 3.1. GC-EAD Analysis of Cuticular Washes

The male cuticular washes contained nine EAG-active compounds (no. 1–9, Figure 3A), while the female cuticle contained six EAG-active compounds (no. 3, 5–9, Figure 3B). The retention times of six compounds coincided in both male and female washes (Figure 3). All nine EAG-active compounds of the male cuticle induced EAG responses in male antennae and four of them elicited EAG responses in the antennae of females. In female cuticular washes, six EAG-active compounds caused EAG responses in the antennae of males.

### 3.2. Compound Identification

In the cuticular washes of *T. autumnalis*, all nine compounds were identified with carbon backbone chain lengths ranging from C21 to C27. The hydrocarbons were attributed to *n*-alkanes (four compounds), alkenes (three compounds), monomethyl alkanes (one compound), and alkadienes (one compound).

The monoenes were readily characterized by determining the double bond position through EI-MS analysis of their dithiomethyl ethers. A 25-carbon alkadiene was identified as 6,9-pentacosadiene by the interpretation of the mass spectra of its DMDS addition products. When double bonds were separated by one, two, or three methylene groups, tetrathiomethyl addition resulted in four-, five-, or six-membered cyclic thioesters substituted with two alkyl chains, each containing a methylthio group alpha to the ring [12]. Fragmentation of the dialkyl cyclic thioester between carbons 6 and 7 resulted in mass peaks at 131 and 343 m/z. The loss of CH_3_-SH from the 343 ion resulted in a peak at 295 m/z. Likewise, fragmentation between carbons 9 and 10 resulted in mass peaks at 271 and 203 m/z. The subsequent loss of CH_3_-SH from the 203 ion resulted in a peak at 155 m/z (Table 1). The compound 6,9-pentacosadiene was characterized as (*Z*, *E*)-6,9-pentacosadiene, and its isomers were determined by comparison with synthetic standards (Figure 4).

The enantiomer of 3-methylheneicosane was identified based on a comparison of synthetic (*R*)- and (*S*)-structures. Behavioral tests revealed a repellent effect of the (*S*)-enantiomer and an excitatory effect of the (*R*)-enantiomer in males (Table 2). Thus, (*R*)-3-methylheneicosane was considered as the enantiomer present in the cuticular washes.

Thus, in the cuticular washes of *T. autumnalis*, the following nine EAG-active compounds were identified: *n*-heneicosane (21Hy), (*R*)-3-methylheneicosane (*R*3me-21Hy), *n*-tricosane (23Hy), (*Z*)-9-tricosene (*Z*9-23Hy), *n*-pentacosane (25Hy), (*Z*)-9-pentacosene (*Z*9-25Hy), (*Z*)-7-pentacosene (*Z*7-25Hy), (*Z*,*E*)-6,9-pentacosadiene (*Z*6*E*9-25Hy), and *n*-heptacosane (27Hy) (Table 1).

All nine identified compounds of the male cuticle induced EAG responses in male antennae, and four of them (namely *Z*9-23Hy, *Z*9-25Hy, *Z*6*E*9-25Hy, and 27Hy) elicited EAG responses in the antennae of females. In the female cuticular washes, six compounds caused EAG responses (23Hy, 25Hy, 27Hy, *Z*9-25Hy, *Z*7-25Hy, and *Z*6*E*9-25Hy) in the antennae of males.

### 3.3. Relative Abundance of the Compounds

Although all the identified EAG-active compounds were present in the cuticular washes of both females and males, they were found in different ratios. Females contained 24 times more *Z*6*E*9-25Hy and seven to nine times more of other C25 compounds (25Hy; *Z*9-25Hy; *Z*7-25Hy) in their cuticle compared to males (Table 1).

In contrast, shorter carbon-chain compounds prevailed in the male cuticle that contained 26 times more *R*3Me-21Hy, 21 times more *Z*9-23Hy, and 13 times more 21Hy than the female cuticle (Table 1).

Two compounds, both saturated hydrocarbons (23Hy and 27Hy), were present in approximately equal abundance in the cuticles of females and males (Table 1).

### 3.4. Male Behavioral Responses to Conspecific CHCs

Male behavioral responses to female cuticular washes were revealed based on two criteria. Males spent significantly more time at the stimulus zone containing female cuticular washes than in the zone with the control (Table 2, Z = 2.1, *p* = 0.036, Figure 5). Additionally, males were significantly faster in choosing the stimulus zone treated with female washes than the zone with the control (Table 2, Z = 2.1, *p* = 0.036). There were no significant differences in the number of visits to the stimulus zone, total distance moved, or male velocity when female cuticular washes were present or absent (control) (Table 2).

When testing male responses to five single CHCs, the following compounds abundant in the female cuticle were found to be the most attractive: 25Hy, *Z*9-25Hy, and *Z*6*E*9-25Hy. Males spent significantly more time at the stimulus zone containing any of these synthetic compounds compared to the control and took significantly less time to make their first visit to the stimulus zone (Table 2). Additionally, two compounds, 25Hy and *Z*6*E*9-25Hy, also increased mobility, movement velocity, and the distance moved (Table 2).

When testing the male behavior response to two single compounds abundant in the male cuticle, statistically significant responses were recorded; *R*3Me-21Hy and *Z*9-23Hy decreased the latency to visit the stimulus zone and increased the distance moved, velocity, and overall mobility (Table 2).

## 4. Discussion

The analysis of the cuticular washes from the female *Tipula autumnalis* revealed that they elicit behavioral responses in males, suggesting the presence of a sex pheromone. Both female and male cuticular washes contained nine EAG-active compounds, identified as hydrocarbons. Although these compounds were found in the cuticles of both sexes, their relative ratios differed. This phenomenon, where cuticular hydrocarbons (CHCs) are not strictly sex-specific, is not uncommon in Diptera [13,14].

This study provides the first evidence of CHCs being detected by olfactory receptors on the antennae of crane flies (Tipulidae). Male antennae of *T. autumnalis* responded to nine CHCs, with the strongest EAG responses elicited by *Z*6*E*9-25Hy, *Z*9-25Hy, and 25Hy. These compounds were present in significantly higher amounts on the female cuticle compared to the male. In behavioral assays, these three compounds were attractive to males, suggesting that they play a critical role in female recognition and function as sex pheromone components. This represents the first data on cuticle-derived compounds eliciting behavioral responses in conspecific males within the Tipulidae family.

Although these three compounds are components of the sex pheromone, they do not induce copulatory attempts in males. Observations indicated that both the cuticular washes and the individual synthetic compounds caused courtship behaviors, such as fanning, grabbing, and mounting. However, no copulatory attempts were observed, even when female dummies treated with cuticular washes or synthetic compounds were used. This lack of copulatory response may be due to the absence of other possibly still missing compounds or certain behavioral signals typically provided by live females. Male *T. autumnalis* appear to require specific signals, such as abdominal elevation or rhythmic up-and-down abdominal movements, to progress to copulation. Studies of *T. oleracea* courtship pattern suggest a complex sequence of reciprocal stimulus–response interactions. For instance, when males exhibit a grabbing response, females respond by lifting their legs [15]. It is possible that similar behaviors are required in *T. autumnalis* to complete the courtship sequence and achieve copulation.

Females contained *Z*7-25Hy in higher amounts than males, though at much lower levels compared to the other compounds discussed above. Additionally, the male EAG response to this CHC was minimal and as result, it was not selected for behavioral testing. Three saturated hydrocarbons (21Hy, 23Hy, and 27Hy) were not behaviorally tested, as when searching for sex pheromone components in females, two compounds, 23Hy and 27Hy, were found in approximately equal amounts in the cuticle of females and males, making it unlikely that they could function as sex-recognition signals. 21Hy was skipped because it was more abundant in males than in females.

Male antennae responded to CHCs 21Hy, *R*3Me-21Hy, and *Z*9-23Hy, which were more abundant in the male cuticle. Notably, *Z*9-23Hy also elicited responses in female antennae, suggesting that *Z*9-23Hy may function as a semiochemical important to both sexes, potentially playing a role in excitation. CHCs 21Hy and *R*3Me-21Hy, on the other hand, appear to be involved in interactions in males. Bioassays conducted in cages confirmed the excitatory role of *R*3me-21Hy and *Z*9-23Hy in males. Under natural field conditions, where insects can move and fly freely, these compounds may stimulate behaviors such as lekking, a phenomenon observed in some dipterans and often associated with lekking or swarming pheromones [16,17].

Two saturated CHCs, 23Hy and 27Hy, were detected in similar quantities in both male and female cuticles. Compound 27Hy was detected by olfactory receptors in both sexes, while 23Hy was perceived only by males. However, the behavioral roles of these compounds remain unclear.

All olfactory-active CHCs identified in *T. autumnalis*, except dienes, are also found in the cuticle of other Diptera species, including mosquitoes, flies, and fruit flies. However, information on their biological function is generally limited and primarily confined to chemical analyses. The following discussion explores their known roles in behavior.

Compounds such as 21Hy and 23Hy are known to influence the behavior of females in a few species, often acting as oviposition-attractant pheromones. For example, 23Hy attracts fertilized females of *Aedes aegypti* to oviposition sites [18], while 21Hy similarly attracts females of *Musca domestica* (Muscidae) to egg-laying sites [19]. The function of 25Hy remains unknown, but its concentration appears to vary with age, suggesting its potential use as an age marker in certain fly species [20,21]. Compound 27Hy, recognized as a sex pheromone, enhances mating competitiveness in male *Aedes aegypti* and *Anopheles stephensi* mosquitoes [22,23]. Our findings reveal a novel behavioral role for 25Hy, as it elicited responses in *T. autumnalis* males. This suggests a novel function not only specific to this species but for the entire order Diptera.

There are significantly more data available on monoenes. Compounds such as *Z*9-23Hy (commonly known as muscalure [24]) and Z9-25Hy are well documented as sex pheromones in various dipteran species from families such as Muscidae, Faniidae, Drosophilidae, and Tephritidae [25,26,27]. Based on studies of *Z*7-25Hy, it has been hypothesized that this compound also functions as a sex pheromone in *Drosophila montana* [28]. Our findings, demonstrating the attractiveness of *Z*9-25Hy to *T. autumnalis* males, expand the number of species and families (including Tipulidae) in which the biological activity of this compound has been identified.

In contrast, very limited data exist on the biological activity of dienes. For example, 6,9-pentacosadiene has been detected only in a single species, *Eurytoma amygdali* (Hymenoptera), where it functions as a sex pheromone in its (*Z*,*Z)*-isomeric form [29]. In our study, the (*Z*,*E)*-isomer of this diene identified in *T. autumnalis* represents a novel stereostructure of pheromones within the insect order.

The methyl-branched compound, 3-methylheneicosane, is commonly found in the cuticle of many insects [30], including a few dipterans from Muscidae and Tephritidae families [30,31]. It is also a predominant component in the cuticular hydrocarbon mixtures of Hymenoptera [2]. The novelty of our work lies in demonstrating, for the first time, the olfactory activity of this compound in insects. Additionally, we provide the first evaluation of the biological activity of its *R*- and *S*-enantiomers based on responses at both olfactory receptor and behavioral levels.

## 5. Conclusions

The first evidence of the presence of sex pheromones in crane flies (Diptera, Tipulidae) was presented. A mixture of hydrocarbons in the cuticle of wingless *T. autumnalis* females is attractive to conspecific males. At least three compounds are attractive, namely *Z*6*E*9-25Hy, *Z*9-25Hy, and 25Hy. The hydrocarbon composition of the cuticle of males and females is similar but significantly differs in the ratio of components. Chemical communication in crane fly *T. autumnalis* is notably complex. Beyond the established sex pheromone components involved in female–male interactions, it is clear that *T. autumnalis* also employs CHCs for male–male interactions. The role of a few CHCs identified as perceived by the olfactory receptors of *T. autumnalis* remains undetermined at this time.

## Figures and Tables

**Figure 1 insects-16-00024-f001:**
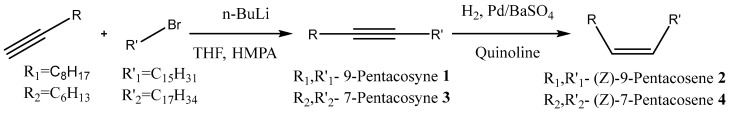
Synthesis scheme of (*Z*)-9-pentacosene and (*Z*)-7-pentacosene.

**Figure 2 insects-16-00024-f002:**
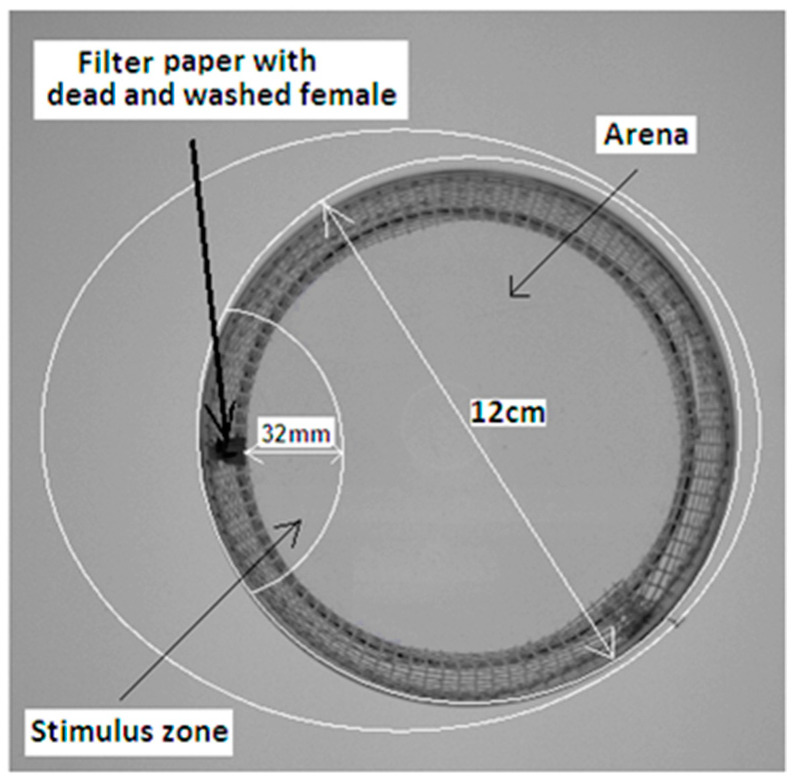
The experimental arena with defined stimulus zone (63 mm in length and 32 mm in width) containing a filter paper (5 × 20 mm) treated with either 10 μL of the stimulus (female cuticular washes or compound) or the control (10 μL of solvent after evaporation).

**Figure 3 insects-16-00024-f003:**
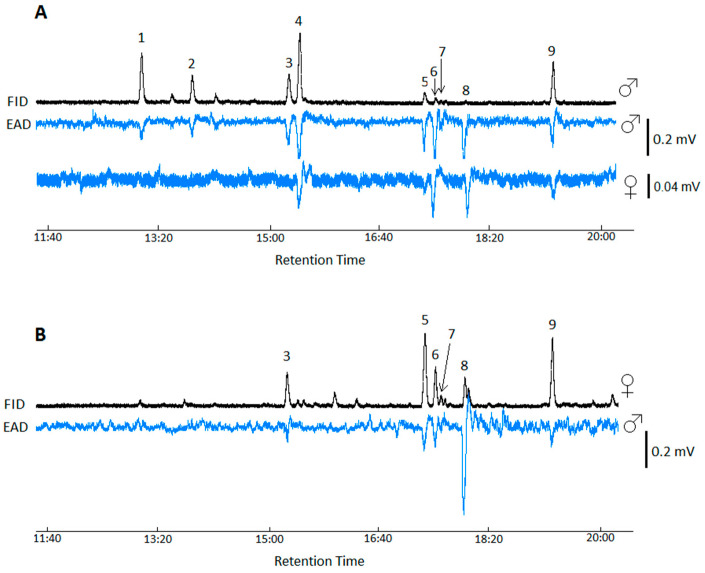
GC-EAD responses of males and females of *Tipula autumnalis* Loew to male (**A**) and female (**B**) cuticular hydrocarbons, namely (1) *n*-heneicosane, (2) (*R*)-3-methylheneicosane, (3) *n*-tricosane, (4) (*Z*)-9-tricosene, (5) *n*-pentacosane, (6) (*Z*)-9-pentacosene, (7) (*Z*)-7-pentacosene, (8) (*Z*, *E*)-6,9-pentacosadiene, and (9) *n*-heptacosane.

**Figure 4 insects-16-00024-f004:**
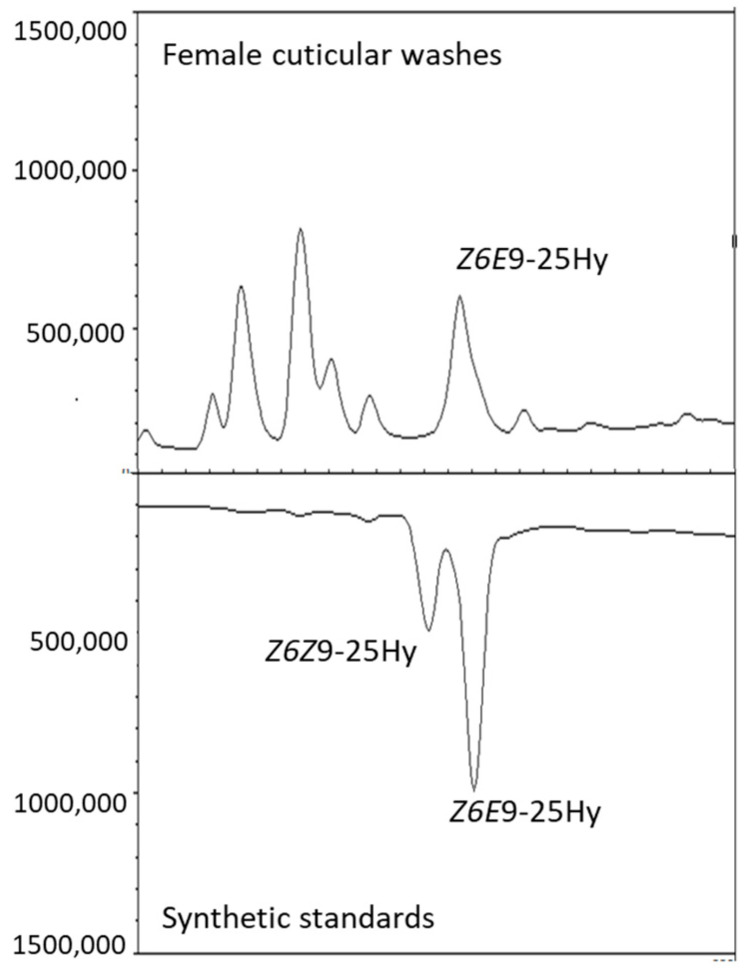
Identification of (*Z*,*E*)-6,9-pentacosadiene in *Tipula autumnalis* Loew female cuticular washes (**top**) based on synthetic standards (**bottom**) on a polar column.

**Figure 5 insects-16-00024-f005:**
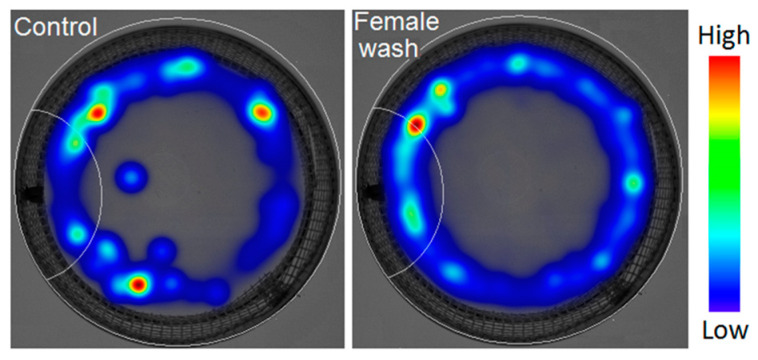
Heatmaps of *Tipula autumnalis* Loew males’ behavior over a 40 min period in the arena illustrate their activity when a female cuticular wash was absent (control) or present (groups of 8 males). The heatmaps visualize the time spent by an insect in a certain position using a color gradient.

**Table 1 insects-16-00024-t001:** EAG-active compounds identified in the cuticular washes of both male and female *Tipula autumnalis* Loew.

Peak No	RI Nonpolar	RI Polar	Compound	Abbreviation	*m/z*		Relative Amount ^1^ (Mean ± SE) (*n* = 4)	
					Untreated	Dimethyl Disulfide Derivative	Female	Male
1	2100	2100	Heneicosane	21Hy	296		1.61 ± 0.02	21.79 ± 0.11
2	2173	2169	(*R*)-3-Methylheneicosane	*R*3Me-21Hy	310, 252/253, 280/281		0.44 ± 0.08	11.55 ± 0.05
3	2300	2300	Tricosane	23Hy	324		13.53 ± 0.06	13.02 ± 0.02
4	2269	2323	(*Z*)-9-Tricosene	*Z*9-23Hy	322	416, 173, 243	1.46 ± 0.01	30.80 ± 0.03
5	2500	2500	Pentacosane	25Hy	352		31.45 ± 0.05	4.28 ± 0.04
6	2471	2515	(*Z*)-9-Pentacosene	Z9-25Hy	350	444, 173, 271	14.17 ± 0.05	1.62 ± 0.08
7	2478	2523	(*Z*)-7-Pentacosene	Z7-25Hy	350	444, 145, 299	3.58 ± 0.05	0.49 ± 0.14
8	2464	2560	(*Z*,*E*)-6,9-Pentacosadiene	Z6*E*9-25Hy	348	442, 131, 155, 203, 271, 295, 343	9.32 ± 0.04	0.38 ± 0.01
9	2700	2700	Heptacosane	27Hy	380		24.44 ± 0.19	16.07 ± 0.18

^1^ Relative amount is expressed as % of the total area of nine peaks.

**Table 2 insects-16-00024-t002:** Behavioral test results of *Tipula autumnalis* Loew males in response to female cuticular wash and its compounds. Behavioral effects are expressed as a percentage relative to control.

Behavioral Criteria	Female Wash	25Hy ^1^	*Z*9-25Hy ^2^	*Z*6*E*9-25Hy ^2^	*Z*9-23Hy ^1^	*R*3Me-21Hy ^1^	*S*3Me-21Hy ^1^
Duration spent in the SZ	159 ± 20 *	151 ± 21 *	221 ± 42 *	235 ± 37 *	112 ± 15	93 ± 14	48± 10 *
Latency to first visit of the SZ	14 ± 7 *	27 ± 18 *	24 ± 9	23 ± 11 *	7 ± 3 *	8 ± 3 *	26 ± 12
Number of visits	171 ± 39	218 ± 26 *	176 ± 34	180 ± 46	161 ± 22	212 ± 32 *	229 ± 70
Distance moved	147 ± 28	226 ± 26 *	134 ± 24	232 ± 39 *	199 ± 31 *	247 ± 35 *	226 ± 56
Velocity	158 ± 32	220 ± 25 *	135 ± 24	230 ± 39 *	185 ± 28 *	246 ± 35 *	235 ± 63
Mobility	136 ± 27	214 ± 24 *	128 ± 21	243 ± 38 *	215 ± 35 *	264 ± 37 *	201 ± 40

* Statistically significant difference between the treatment and control (*p* < 0.05, Mann–Whitney U test). ^1^ Concentration 0.1 mg/mL; ^2^ concentration 0.01 mg/mL. SZ—stimulus zone.

## Data Availability

The raw data supporting the conclusions of this article will be made available by the authors on request.

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
