# Peer review of "Pheromones in Crane Flies: Behaviorally Active Cuticular Compounds in Tipula autumnalis Loew (Diptera: Tipulidae)"

_insects, 2024, doi:10.3390/insects16010024_

Round 1
Reviewer 1 Report
Comments and Suggestions for Authors
See attach

The English quality is OK, but I recommend editing of the revised version by an English native expert.
Author Response
Please find details in the file attached.

Reviewer 2 Report
Comments and Suggestions for Authors
General Comments
The authors described pheromones in crane flies, and the experiments were performed logically for identification. However, some points in the description of the EAG and behavioral experiments require amendment.
- Line 152: Specify for which compound the NMR data was provided.
- Lines 247-251: Provide a detailed description of the EAG results, explicitly identifying which compounds from male extracts elicited responses in both males and females, and which compounds from female extracts elicited responses in males.
- Behavioral tests:
- The compounds 21Hy, Z9-23Hy, Z7-25Hy, and 27Hy were not tested. While the reason for not testing Z7-25Hy was provided, the reasons for not testing the other compounds were not mentioned. Please explain why these compounds were excluded from the behavioral tests.
- Amount used in behavioral tests: Although the ratio of 25Hy:Z9-25Hy:Z6,E9-25Hy was 1:0.5:0.3 for females, the amounts used for the behavioral test were 0.1:0.01:0.01 mg/L. Clarify why these specific amounts were chosen.
- Testing pheromonal activity: Typically, when testing the pheromonal activity of hydrocarbons (CHs), experiments are conducted by removing one component at a time from the combined mixture. If feasible, conduct these experiments to identify the critical pheromone components.
Author Response
Please find the responses in the attached file.
